

# How C: N: P stoichiometry in soils and carbon distribution in plants respond to forest age in a *Pinus tabuliformis* plantation in the mountainous area of eastern Liaoning Province, China

Lijiao Wang[1,2], Xin Jing[1,2], Jincheng Han[1,2], Lei Yu[1,2], Yutao Wang[1,2], Ping Liu[1,2,3]

[1] College of Forestry, Shenyang Agricultural University, Shenyang, Liaoning, China
[2] Key Laboratory of Tree Genetics, Breeding and Cultivation in Liaoning Province, China, Shenyang, China
[3] Engineering Technology Research Center of Chinese Pine of National Forestry and Grassland Administration, Shenyang, China

Corresponding author
Ping Liu, lp_79@163.com

## ABSTRACT

Carbon distribution in plants and ecological stoichiometry in soils are important indicators of element cycling and ecosystem stability. In this study, five forest ages, young forest (YF), middle-aged forest (MAF), near-mature forest (NMF), mature forest (MF), and over-mature forest (OMF) in a *Pinus tabuliformis* plantation were chosen to illustrate interactions among the C: N: P stoichiometry in soils and carbon distribution in plants, in the mountainous area of eastern Liaoning, China. Carbon content was highest in the leaves of MAF (505.90 g·kg$^{-1}$) and NMF (509.00 g·kg$^{-1}$) and the trunks of YF (503.72 g·kg$^{-1}$), MF (509.73 g·kg$^{-1}$), and OMF (504.90 g·kg$^{-1}$), and was lowest in the branches over the entire life cycle of the aboveground components (335.00 g·kg$^{-1}$). The carbon content of the fine roots decreased with soil layer depth. In YF, MAF, and NMF carbon content of fine roots at 0.5 m was always higher than that of fine roots at 1 m; however, it was the opposite in MF and OMF. The carbon content of the leaves changed with forest age; however, carbon content of branches, trunks and fine roots did not change significantly. Soil total carbon (TC), total nitrogen (TN), total phosphorus (TP), and available phosphorus (AP) content was highest in the OMF. Soil TC, TN and AP content, and TC: TN, TC: TP and TN: TP ratio decreased with increasing soil depth. Soil TC, TN, and TP content had a significant effect on the carbon content of fine roots ($p < 0.05$). The leaf carbon content and soil element content changed obviously with forest age, and the soil TN, TP and AP increased, which might reduce the carbon content allocation of fine roots.

## INTRODUCTION

The forest ecosystem is an important part of the terrestrial ecosystem and the Earth's biosphere. Such ecosystems play an important role in regulating and maintaining the

Earth's ecosystems (*Shao et al., 2017*). Forest ecosystems contain plenty of carbon elements, making an important contribution to the global carbon sink (*Dixon et al., 1994*). Forest ecosystem carbon storage accounts for approximately 56% of the terrestrial ecosystem carbon storage, of which forest vegetation carbon storage accounts for more than 80% of the global vegetation carbon storage, and the forest soil carbon pool accounts for more than 40% of the global soil carbon pool (*Brown, Schroeder & Kern, 1999*; *Houghton et al., 2001*; *Yang et al., 2019*). Various human activities, such as burning fossil fuels (*Haimson & Ennis, 2004*) and deforestation, have led to the continual increase of $CO_2$ gas, forming the greenhouse effect and severely affecting the ecological environment, and by increasing forest carbon storage, greenhouse gas emissions from forests can be avoided, and climate change can be mitigated, thus protecting the ecological environment of the earth (*Liu et al., 2015a*; *Liu et al., 2015b*). The forest ecosystem carbon cycle is closely related to the dynamic balance of vegetation and soil carbon storage (*Busse et al., 2009*). Soil is an important factor in terrestrial ecosystems (*Gusewell, 2004*). As key components of forest ecosystems, trees harbor approximately 42% carbon in their live biomass (*McDowell et al., 2020*; *Pan et al., 2011*). Carbon, nitrogen, and phosphorus cycles account for the transfer of nutrients between plants and soil. Carbon is a key building block of structural materials, and nitrogen and phosphorus are major limiting elements in terrestrial natural ecosystems (*Zhao et al., 2017*). The soil total phosphorus (TP) content is very low (approximately 0.02–2%), and China's soil survey estimated that nearly 70% of its soil is deficient in phosphorus (*He et al., 2020*). The level of soil nutrient content directly affects the growth and development of trees, level of productivity, and ecological function (*Wang et al., 2020*). The three nutrient elements interact with each other during cycling, and nitrogen and phosphorus affect carbon fixation in soil (*Han et al., 2005*).

At present, in relevant local and global studies, the influence of forest age on forest carbon content was found to be particularly important (*Wei & Man, 2019*). The results of related studies (*Hu et al., 2014*; *Ming et al., 2014*; *Zhu et al., 2017*) have shown that within a specific range of forest ages, the carbon storage of vegetation increases with increasing forest age. Mountainous areas in eastern Liaoning Province have a temperate monsoon climate, with dry winter and rainy summer, which are relatively humid and suitable for the growth of *Pinus tabuliformis*. However, most of the research has been concentrated in the subtropical areas of China, and there are few studies on the eastern area of Liaoning Province. Most scholars have studied the relationships between soil, leaves, and microorganisms, such as *Deng et al. (2019)*. The relationship between soil and litter in pine forests compared to the soil stoichiometric characteristics among different tree species were studied by scholars (*Jiang et al., 2016*; *Wang & Zheng, 2020*; *Zhang et al., 2018*; *Qi et al., 2020*). There are many studies on carbon content in the leaves of *Pinus tabuliformis* (*Yang et al., 2020*; *Yan et al., 2021*; *Song, Zhou & Zhang, 2021*); however, few studies have been undertaken on studying other organs. There are many studies on ecological stoichiometry characteristics (*Wu, 2020*; *Liu et al., 2020*; *Jing et al., 2018*); however, few analyses have been conducted on the relationship between underground and aboveground areas. Therefore, studying only the carbon content of some organs or studying only the indicators of a period of growth may

not reflect the forest carbon sequestration potential and the relationship between plants and the soil in a region.

The local tree species and main pioneer tree for afforestation in Liaoning Province is *Pinus tabuliformis*, with a total area of approximately 700,000 hm$^2$ and a cumulative stock volume of about 36 million m$^3$ (*Liu et al., 2019*). These species are an important part of the forest re-sources and play very important ecological roles in maintaining the forest ecological balance, saving water resources, and protecting the diversity of wild animals and plants in the eastern mountainous area of Liaoning Province. To implement successful forest management policies, a scientific basis is required (*Tewari, 2016*). Therefore, the present study used *P. tabuliformis* plantation in Fushun County as the research object to determine the content of carbon in different organs (leaves, branches, trunks, and fine roots) and the total carbon (TC), total nitrogen (TN), total phosphorus (TP), and available phosphorus (AP) content in the soil over the entire life cycle of a *P. tabuliformis* plantation. The objectives of this study were to test the following three hypotheses: (1) assuming that the carbon distribution of *P. tabuliformis* plantations was age-dependent, the carbon content of the plant organs would change with the different needs of the plants at different growth stages; (2) assuming that the C: N: P stoichiometric characteristics of the soil are highly dependent on stand age and soil depth, the passing of time and change in plant growth requirements would mean that the carbon, nitrogen, and phosphorus elements in all soil layers would change and accumulate to different degrees; and (3) from the growth cycle of YF to OMF, soil C: N: P stoichiometry would significantly affect plant carbon distribution.

## MATERIALS & METHODS

### Study site and experimental design

The mountainous area of eastern Liaoning Province, China, belongs to the temperate monsoon climate zone, with a long cold winter and short rainy summer. The average annual precipitation is 700–850 mm, the average annual evaporation is 925–1,284 mm, the average annual temperature is 4-11 °C, and the frost-free period is 120–139 d long. This region belongs to the extension area of the Changbai Mountains, with an altitude of 200–500 m, and a few peaks exceeding 1,000 m. The soil is dominated by dark brown loam, generally acidic or neutral loam.

In the present study, *P. tabuliformis* plantation sample plots with similar density and different forest ages were established in the Magu Forest Farm of Fushun County (College of Forestry of Shenyang Agricultural University and Magu Forest Farm, Forestry Department of Fushun Mining Group Co. Ltd approval), Liaoning Province, and included five age classes: young forest (YF), middle-aged forest (MAF), near-mature forest (NMF), mature forest (MF), and over-mature forest (OMF). Each age level had three sample plots, for a total of 15 permanent sample plots. The area of each sample plot was 0.06 hm$^2$ (20 m × 30 m). Two trees representing the average tree height and diameter at breast height (DBH) in each sample plot were selected as standard trees. The DBH, tree height, height under the branch of *P. tabuliformis*, stand density, forest age, and other stand indices were

**Table 1  Basic information of sample trees.**

| Age class | Mean Age (a) | Mean DBH (cm) | Mean tree height (m) | Mean Stand density (tree hm$^{-2}$) |
|---|---|---|---|---|
| Young forest | 10 | 6.3 | 3.8 | 1600 |
| Middle-aged forest | 28 | 18.0 | 10.7 | 1255 |
| Near-mature forest | 31 | 20.3 | 12.3 | 1089 |
| Mature forest | 51 | 20.7 | 12.6 | 900 |
| Over-mature forest | 69 | 27.5 | 15.1 | 563 |

measured using a DBH ruler, tape, and altimeter. The basic data of the samples are shown in Table 1.

## Aboveground samples

The samples were collected in October 2019. Two standard wood plants were selected from each sample plot and a core was drilled through the standard wood from the DBH along a north-south direction with a growth cone, for a total of 30 cores. The standard wood canopy was divided into three layers: upper, middle, and lower. Two standard branches were selected from each layer, for a total of six standard branches. Each standard branch intercepted one sample branch, and one sample leaf, which were saved in a numbered envelope. A total of 360 samples were collected from 30 standard trees with 180 branch samples and 180 leaf samples. The branch and core samples were transported to the laboratory for drying at 80 °C to a constant weight. The leaves were cured at 105 °C for 30 min, and then dried to a constant weight at 65 °C. The aboveground part totaled 390 samples.

## Fine roots and soil samples

Soil and fine root samples were collected in October 2019. The underground part of the standard wood was sampled by using the soil drilling method. A cross section of 1 m × 0.5 m × 0.6 m in length × width × depth was dug at the base of each standard wood, and 180 fine roots samples were taken from three soil layers at different depths (0–20 cm, 20–40 cm, and 40–60 cm), 0.5m and 1 m from the base of the trunk. And 45 soil samples at three different depths (0–20 cm, 20–40 cm, and 40–60 cm) from each plot were collected. All fine roots (<2 mm) were removed from soil and the soil samples and fine roots that had rocks and debris removed were transported to the laboratory. After the fine roots were washed and air-dried naturally, they were placed in an oven, dehydrated at 105 °C for 30 min, and dried to a constant weight at 65 °C. Soil samples were naturally air-dried.

## Elements in *Pinus tabuliformis* and soil samples

Before determining the element content, the samples of leaves, branches, wood cores, fine roots (same distance, same soil layer) from the same sample plot were mixed respectively to form composite samples. The composite samples of wood cores, sample branches, sample leaves, and fine roots were ground and screened with a 60-mesh sieve, whereas the soil
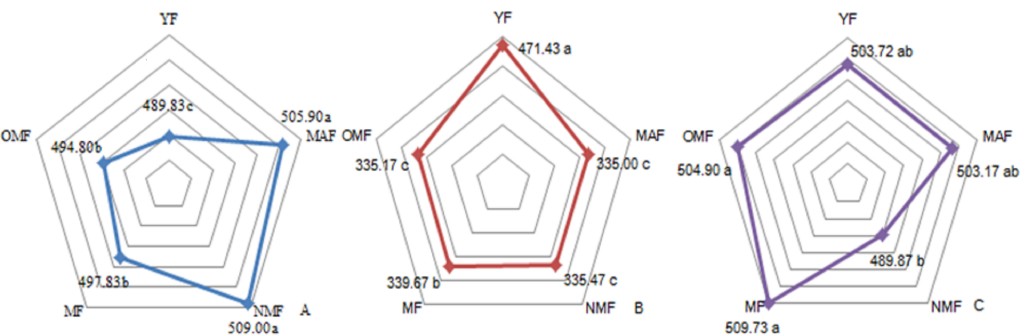

**Figure 1 Carbon content of leaf (A), branch (B) and trunk (C).** Different lowercase letters indicate significant differences among different forest ages ($p < 0.05$). YF, Young forest; MAF, Middle-aged forest; NMF, Near-mature forest; MF, Mature forest; OMF, Over-mature forest.

composite samples were air-dried and screened with a 100-mesh sieve. All samples were then placed into self-sealing bags, numbered and sealed, and determination.

The TC of plant samples and TC and TN of soil samples were determined using an elemental analyzer (Vario EL III; Elementar. Langenselbold, Germany). Soil TP was determined using the $HCLO_4$-$H_2SO_4$-molybdenum-antimony colorimetric method. The soil AP was determined with the molybdenum-antimony colorimetric method.

### Statistical analysis

Data processing was performed using SPSS 22.0 software (SPSS, Inc., Chicago, IL, USA). The differences in TC content in the different organs and soil TC, TN, TP, AP and its ratio throughout the entire life cycle (from YF to OMF) were examined using single factor variance analysis. Duncan's multiple comparison method was used for significance analysis ($p < 0.05$). Pearson's correlation analysis was used to analyze the correlation between TC, TN, TP, AP and the ratio of the surface soil (0–20 cm) and the carbon content of different organs over the entire life cycle of the *P. tabuliformis* plantation.

## RESULTS

### Carbon distribution of *Pinus tabuliformis* plantation
#### *Aboveground carbon distribution*

The carbon content over the entire life cycle of the *P. tabuliformi*s plantation was 489.83–509.00 g·kg$^{-1}$ in the leaves (Fig. 1A), 335.00–471.43 g·kg$^{-1}$ in the branches (Fig. 1B), and 489.87–509.73 g·kg$^{-1}$ in the trunks (Fig. 1C). With a change in forest age, the carbon content of the leaves gradually increased and then decreased, reaching a maximum in NMF. There were significant differences between the carbon content in forest ages ($p < 0.05$), except in MAF and NMF, and MF and OMF (Fig. 1A). The carbon content of the branches decreased from YF to MAF, and decreased after a slight increase in MF. There were significant differences between the carbon content in forest ages ($p < 0.05$), except in MAF, NMF, and OMF (Fig. 1B). The carbon content of the trunks only in the NMF showed significant differences between MF and OMF ($p < 0.05$) (Fig. 1C).

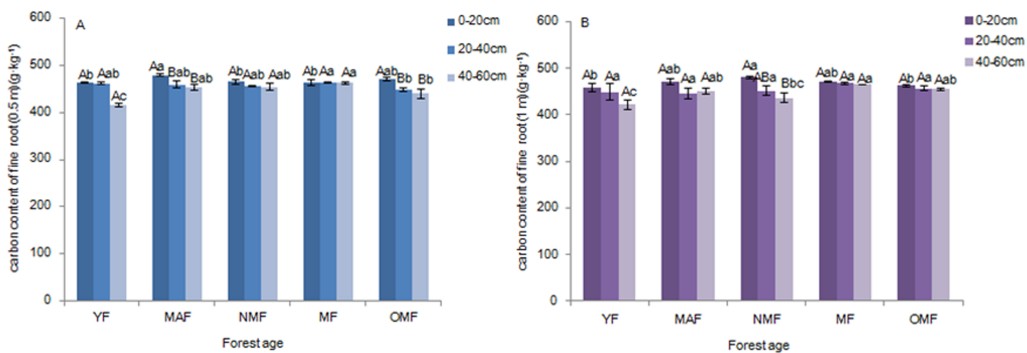

**Figure 2** **Carbon content of fine roots (0.5 m) (A) and fine roots (1 m) (B).** Different uppercase letters in the same forest age indicate significant differences between different soil layers, while different lowercase letters in the same soil layer indicate significant differences between different forest ages ($p < 0.05$). YF, Young forest; MAF, Middle-aged forest; NMF, Near-mature forest; MF, Mature forest; OMF, Overmature forest.

### Underground carbon distribution

The carbon content of fine roots ranged from 415.20 to 480.37 g·kg$^{-1}$ over the entire life cycle of the *P. tabuliformis* plantation. There were no significant differences among the carbon content of fine roots (0.5 m) in the soil layers of YF, NMF, and MF. There were significant differences between the 0–20 cm soil layer ($p < 0.05$) and other soil layers in the MAF and OMF age groups ($p < 0.05$) (Fig. 2A). In the 0–20 cm soil layer, the carbon content of fine roots (0.5 m) in MAF and YF, NMF and OMF were significantly different ($p < 0.05$). In the 20–40 cm soil layer, there was significant difference among MF and OMF ($p < 0.05$). In the 40–60 cm soil layer, YF, MF, and OMF showed significant differences ($p < 0.05$). However, for the carbon content of fine roots (1 m) in all soil layers of different forest ages, only the 40–60 cm and 0–20 cm layers in NMF showed significant differences ($p < 0.05$) (Fig. 2B). In the 0–20 cm soil layer, there were significant differences between NMF, YF, and OMF ($p < 0.05$). In the 20–40 cm soil layer, there was no significant difference among the different forest ages. In the 40–60 cm soil layer, the carbon content of fine roots (1 m) in the YF was significantly different from that in MAF, MF, and OMF ($p < 0.05$), and between NMF and MF ($p < 0.05$). The results showed that the carbon content of fine roots decreased with soil layer depth, and there was a trend of first increasing and then decreasing with an increase in forest age.

### TC distribution

In the YF, the order of carbon content of the different organs was trunks > leaves > branches > fine roots; in MAF and NMF the order was leaves > trunks > fine roots > branches, and in MF and OMF the order was trunks > leaves > fine roots > branches (Table 2). The carbon content of the aboveground parts was significantly different from that of fine roots over the entire life cycle of the *P. tabuliformis* plantation. The carbon content of the leaves and trunks was significantly different from that of the branches, except in the YF. However, there was no significant difference between the carbon content of fine roots in the same forest age. The carbon content of the branches was the lowest

Wang et al. (2021), PeerJ, DOI 10.7717/peerj.11873

**Table 2  Carbon distribution over the life cycle of *Pinus tabuliformis* plantation.**

| Forest Age | Aboveground Carbon Content | | | Underground Carbon Content | | | | | |
| --- | --- | --- | --- | --- | --- | --- | --- | --- | --- |
| | Leaf (g·kg⁻¹) | Branch (g·kg⁻¹) | Trunk (g·kg⁻¹) | Fine Roots (0.5m g·kg⁻¹) | | | Fine Roots (1m g·kg⁻¹) | | |
| | | | | 0–20 cm | 20–40 cm | 40–60 cm | 0–20 cm | 20–40 cm | 40–60 cm |
| YF | 489.83 ± 1.05 AB | 471.43 ± 2.07 BC | 503.72 ± 6.43 A | 463.83 ± 2.38 CD | 461.88 ± 4.07 CD | 415.20 ± 5.66 DE | 458.20 ± 14.87 CD | 448.85 ± 31.74 CDE | 422.32 ± 18.50 E |
| MAF | 505.90 ± 2.72 A | 335.00 ± 1.59 E | 503.17 ± 10.17 A | 479.63 ± 6.18 B | 458.77 ± 12.65 CD | 453.37 ± 9.86 CD | 471.27 ± 12.33 BC | 445.40 ± 19.89 D | 451.20 ± 10.21CD |
| NMF | 509.00 ± 3.72 A | 335.47 ± 2.57 F | 489.87 ± 2.38 AB | 465.13 ± 18.76 CD | 456.77 ± 2.21 DE | 453.80 ± 12.90 DE | 480.37 ± 4.71 BC | 451.93 ± 15.85 DE | 436.73 ± 19.09 E |
| MF | 497.83 ± 3.62 B | 339.67 ± 1.93 D | 509.73 ± 10.03 A | 463.10 ± 0.76 C | 464.43 ± 2.00 C | 462.33 ± 3.81 C | 471.60 ± 2.42 C | 468.67 ± 4.45C | 465.97 ± 0.29 C |
| OMF | 494.80 ± 1.11 A | 335.17 ± 1.72 F | 504.90 ± 6.17 A | 471.43 ± 6.26 B | 449.07 ± 6.44 DE | 439.67 ± 15.88 E | 462.57 ± 3.44 BC | 456.93 ± 7.80 CD | 455.80 ± 4.7CD |

**Notes.**

Different uppercase letters indicate significant differences between the carbon content of different organs in the same forest age ($p < 0.05$).

YF, Young forest; MAF, Middle-aged forest; NMF, Near-mature forest; MF, Mature forest; OMF, Over-mature forest.

of the aboveground components, and the carbon content of the fine roots decreased with the soil layer depth. In YF, MAF, and NMF, the carbon content of fine roots at 0.5 m was consistently higher than that of fine roots at 1 m in the same soil layer; meanwhile, the opposite results were found in MF and OMF.

## Soil ecological stoichiometric characteristics
### Soil TC, TN, TP, and AP content

In the surface soil (0–20 cm), the value of soil TC content was varied between 11.25–37.82 g·kg$^{-1}$, the value of soil TN content was varied between 0.69–2.95 g·kg$^{-1}$, the value of soil TP content was varied between 0.28–1.32 g·kg$^{-1}$, and the value of soil AP content was varied between 13.57–18.57 mg·kg$^{-1}$. And in all soil layers, over the entire life cycle of the *P. tabuliformis* plantation, the value of soil TC content was varied between 3.29 and 37.82 g·kg$^{-1}$ (Fig. 3A), the value of soil TN content was varied between 0.22 and 2.95 g·kg$^{-1}$ (Fig. 3B), the value of soil TP content was varied between 0.23 to 1.61 g·kg$^{-1}$ (Fig. 3C), and the value of soil AP content was varied between 13.52–20.99 mg·kg$^{-1}$ (Fig. 3D). In some forest ages, the TC and TN content in the 0–20 cm soil layer was significantly different from those in the other soil layers; however, the TC and TN content in the 20–40 cm and 40–60 cm soil layers were not significantly different over the entire life cycle. The TP and AP content in the soil did not change significantly with soil layer and forest age, except for the AP content in NMF. Therefore, the TC and TN content in the soil decreased with soil layer depth, and that of TC, TN, TP, and AP was the highest in the OMF in each soil layer, except for AP in the 20–40 cm soil layer.

### Stoichiometric ratio of soil TC, TN, and TP

In the surface soil (0–20 cm), soil TC: TN ratio ranged from 12.82 to 18.00, soil TC: TP ratio ranged from 11.01 to 67.84, and soil TN: TP ratio ranged from 0.73 to 3.97. And in all soil layers, over the entire life cycle of the *P. tabuliformis* plantation, soil TC: TN ratio ranged from 10.64 to 18.00 (Fig. 4A), soil TC: TP ratio ranged from 4.01 to 67.84 (Fig. 4B), and soil TN: TP ratio ranged from 0.29 to 3.97 (Fig. 4C). There was no significant difference in soil TC: TN ratio among the different soil layers in the same forest age. In YF, MAF, and NMF, soil TC: TP and soil TN: TP ratio showed significant differences among the soil layers. The soil TC: TN, TC: TP and TN: TP ratio varied with the content of soil TC, TN and TP. So they show different significant differences at different ages. In the same forest age, soil TC: TN, soil TC: TP, and soil TN: TP ratio showed a downward trend with soil layer depth, and soil TC: TP and soil TN: TP ratio first increased and then decreased with an increase in forest age.

## Correlation of soil ecological stoichiometry and its ratio with the carbon content in different organs

There was a significant positive correlation between the carbon content of leaves and soil TN: TP ratio in the YF (Fig. 5A). The carbon content of leaves had a significant positive correlation with soil AP content (Fig. 5B), and the carbon content of trunks had a significant positive correlation with soil TN: TP ratio in MF (Fig. 5C). The results showed that soil stoichiometry and its ratio had a weak correlation with the aboveground carbon content.
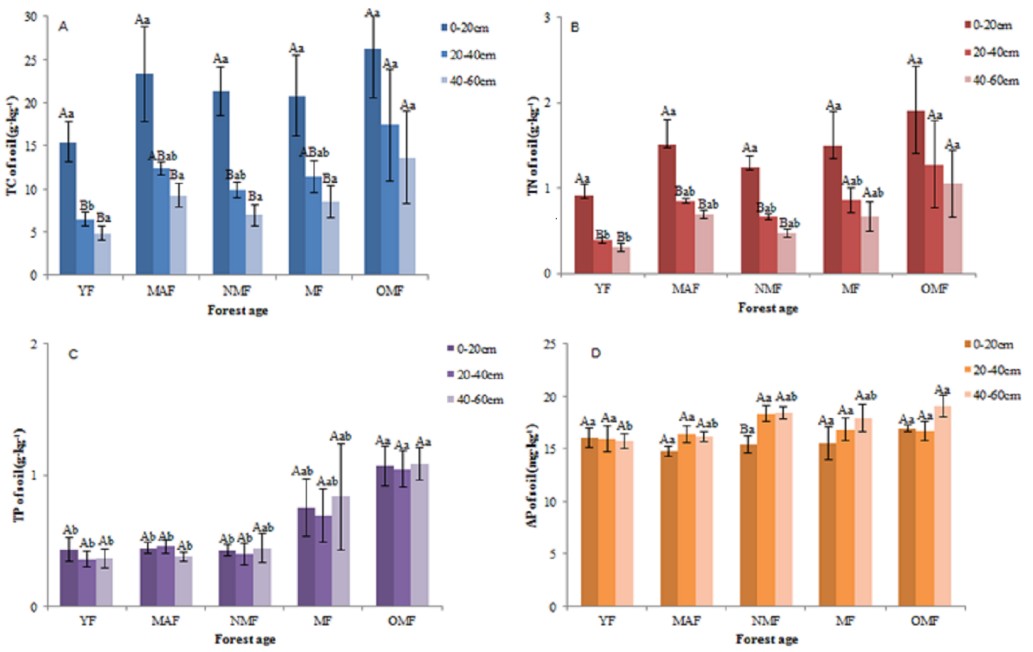

**Figure 3** **Soil TC content (A), soil TN content (B), soil TP content (C) and soil AP content (D).** Different uppercase letters in the same forest age indicate significant differences between different soil layers, while different lowercase letters in the same soil layer indicate significant differences between different forest ages ($p < 0.05$). YF, Young forest; MAF: Middle-aged forest; NMF, Near-mature forest; MF, Mature forest; OMF, Over-mature forest.

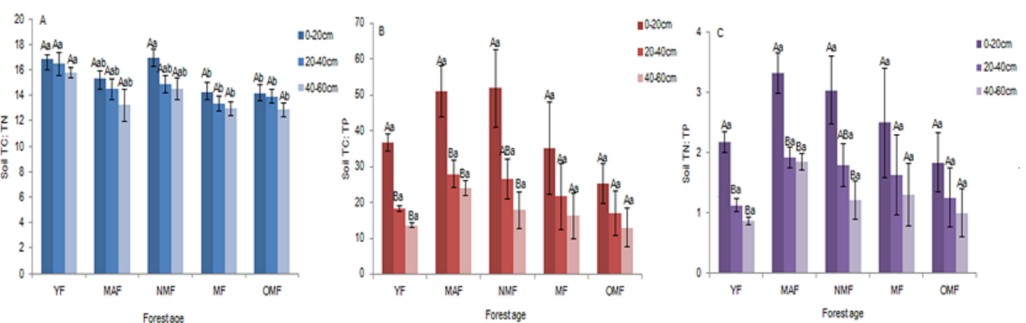

**Figure 4** **Soil TC: TN (A), Soil TC: TP (B) and Soil TN: TP (C).** Different uppercase letters in the same forest age indicate significant differences between different soil layers, while different lowercase letters in the same soil layer indicate significant differences between different forest ages ($p < 0.05$). YF, Young forest; MAF, Middle-aged forest; NMF, Near-mature forest; MF, Mature forest; OMF, Over-mature forest.

Soil TC: TN ratio was significantly positively correlated with the fine roots (0.5 m) in the 20–40 cm soil layer. Soil TN and TP content was significantly negatively correlated with the fine roots (1 m) in the 0–20 cm soil layer, and TC: TP ratio was significantly positively correlated with the fine roots in the YF (Figs. 6A–6D). The soil AP content was significantly negatively correlated with the fine roots (0.5 m) in the 20–40 cm soil layer in the MAF (Fig. 6E). Soil TC content was significantly negatively correlated with the fine

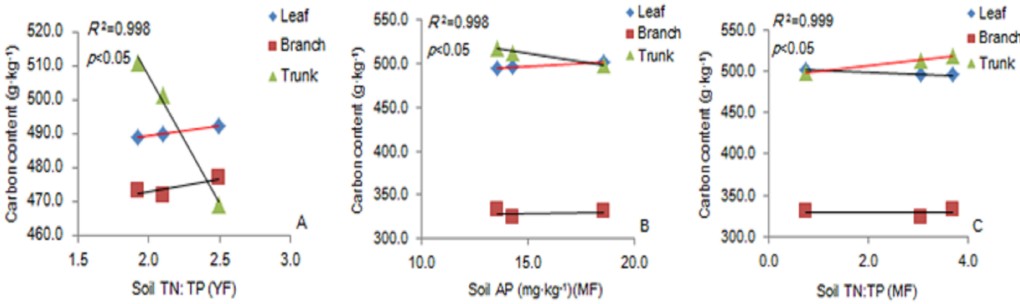

**Figure 5** **The linear relationship between the components of YF (A) and MF (B, C).** The correlation ($p$ <0. 05) is indicated by a red line. YF, Young forest; MF, Mature forest.

roots (0.5 m) in the 40–60 cm soil layer (Fig. 6F). The soil TN content soil was significantly negatively correlated with the fine roots (0.5 m) in the 20–40 cm soil layer in the NMF (Fig. 6G). Soil TC: TP ratio was significantly positively correlated with the fine roots (0.5 m) in the 40–60 cm soil layer in the MF (Fig. 6H), and the soil TN content was significantly negatively correlated with fine roots (1 m) in the 20–40 cm soil layer in the OMF (Fig. 6I). Therefore, the soil stoichiometry and its ratio strongly correlated with the carbon content of fine roots.

## DISCUSSION

### Carbon distributions over the life cycle of *Pinus tabuliformis* plantation

The result of this part of the study is consistent with our first hypothesis. The carbon content in the leaves of *P. tabuliformis* was obviously higher than that (464 g·kg$^{-1}$) of the 492 terrestrial plants worldwide studied by *Elser et al. (2000)*, indicating a high content of organic compounds in *P. tabuliformis* leaves ((*Wang & Zheng, 2018)*. The carbon content of leaves, branches, and trunks was obviously different from that reported by *Liu et al. (2015a)*; *Liu et al. (2015b)* and which may be due to the comprehensive influence of different factors, including sampling time, forest age, site conditions, climate, and environment of the sampling site (*Wang & Zheng, 2018*). The carbon content of the branches decreased with an increase in forest age, whereas the carbon content of the trunks did not change significantly with an increase in forest age, which was consistent with that reported by *Zhang et al. (2018)*. This may be due to the difference in the carbon synthesis rate and distribution strategy in the leaves and branches at different ages (*Lei, 2019*). The carbon content in the leaves and branches of *P. tabuliformis* with an increase in forest age, indicating that the fixed carbon content decreased with the growth of plant organs, and the change in content was easily affected by a variety of factors, including sampling time, forest age, and soil physical and chemical properties (*Zhang et al., 2013*).

During the early growth stage of trees, with an increase in growth rate, the carbon distribution of fine roots closer to the trunk of *P. tabuliformis* increased, and to meet the needs of tree growth, the fine roots absorbed water and nutrients continuously. However,

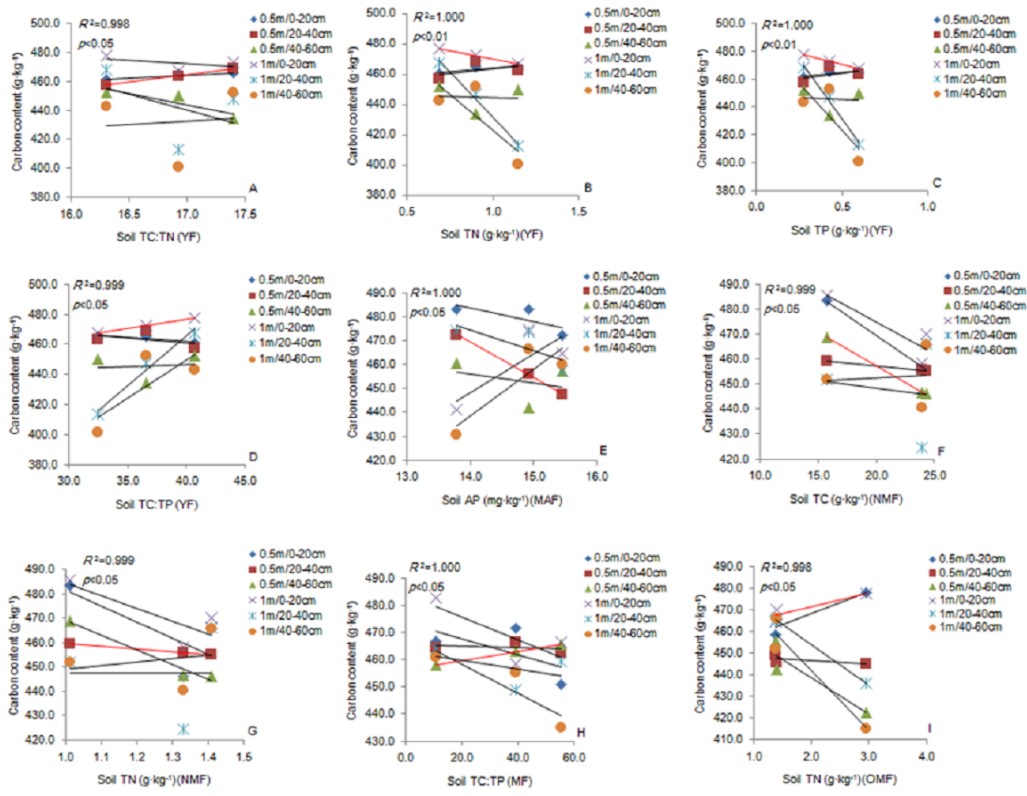

**Figure 6** **The linear relationship between the components of YF (A, B, C, D), MAF (E), NMF (F, G), MF (H) and OMF (I).** The correlation ($p < 0.05$) is indicated by a red line. YF, Young forest; MAF, Middle-aged forest; NMF, Near-mature forest; MF, Mature forest; OMF, Over-mature forest.

during the later growth stage of trees, when the growth rate reached the maximum, the growth rate slowed down or stopped, and the roots mainly grew and spread around to expand the absorption area of nutrients. Therefore, during the later growth stage of trees, the fine roots farther away from the trunk had higher carbon content.

### Soil ecological stoichiometry characteristics over the life cycle of *Pinus tabuliformis* plantation

The accumulations of soil TC, TN, and AP are long-term process, and the soil TP content had the most obvious accumulation trend with an increase in forest age. This is consistent with our second hypothesis. The present study was consistent with the previous research have also found that the TC, TN, and TP content of soil in OMF was the highest (*Zhao et al., 2012*).The results of the present study also showed that the surface layer (0–20 cm) had the highest TC and TN content, showing the phenomenon of "surface accumulation," which is consistent with the results of previous studies and might be related to the high activity of soil surface soil microorganisms (*Wang & Zheng, 2018*). The surface soil TN content (2.95 g·kg$^{-1}$) was higher than the national average total nitrogen content (1.88 g·kg$^{-1}$), indicating that the *P. tabuliformis* plantation in this area had a certain effect on increasing the soil nitrogen content.

In the present study, the soil TP content (1.61 g·kg$^{-1}$) was lower than the global average (2.8 g·kg$^{-1}$) (*Ren et al., 2007*), which is consistent with the fact that the soil TP content in China is generally lower than the global level (*Yang et al., 2014*). It may be related to environmental conditions, weathering, and soil erosion in the mountainous areas of eastern Liaoning Province. The results was consistent with the previous research have also found that there was no significant difference in the TP content among all soil layers (*Wei & Shao, 2007*). This was mainly due to the different carbon, nitrogen, and phosphorus sources (*Wang & Yu, 2008*). Carbon and nitrogen elements accumulate on the soil surface, migrate downward by leaching, and are affected by plant absorption and utilization. Phosphorus is mainly affected by the weathering of the soil parent material. Moreover, phosphorus is a sedimentary mineral that migrates relatively little in the soil, and is evenly distributed throughout the entire soil layer (*Cui, Cao & Chen, 2015*). The weathering of rocks and minerals is a stable and lengthy process, which is a major source of soil AP. Therefore, their spatial variability in the soil is small and changes in the soil layer depths are not significant (*Liu et al., 2010*).

In general, the soil TC: TN ratio is inversely proportional to the decomposition rate of soil organic matter (*Majdi & Ohrvik, 2004*). The soil TC: TN ratio (18.00) in the present study area was higher than the Chinese soil TC: TN ratio (10.1–12.1) and the global average (13.33). Therefore, the decomposition rate of organic matter and mineralization in the study area were relatively slow. However, except for NMF in this study, soil TC: TN ratio in other aged *P. tabuliformis* plantations decreased with forest age, indicating that with the increase of forest age, the mineralization rate of organic matter increased, and the demand for soil nutrients gradually increased. If the soil TC: TP ratio is relatively low, it is conducive to the release of nutrients from microorganisms in the process of organic matter decomposition and to promote the increase of available phosphorus in the soil. On the contrary, when soil TC: TP ratio is relatively high, there will be limited phosphorus in the process of decomposition of organic matter by microorganisms. Therefore, there will be competition with plants for soil inorganic phosphorus, which is not conducive to the growth of plants and the increase of NPP (*Wang et al., 2014*). In the present study, soil TC: TP ratio (67.84) was higher than the average value of 61 in China, indicating that microorganisms and plants in the surface soil layer of the study area compete for phosphorus, which is not conducive to growth (*Ning, 2020*). In the present study, soil TN: TP ratio (3.97) was lower than the mean value of soil TN: TP ratio in China (5.2), and the mean value increasing at the beginning and then decreasing with the change in forest age, but the difference was not statistically significant. The reason may be that with the growth of *P. tabuliformis*, the contents of nutrient elements in soil decreased to different degrees, and the demand for phosphorus was stronger than that for carbon and nitrogen. With the increase in forest age, the uptake of nutrients slows down, and the return of nutrients from the litter supplements the soil. In the present study, soil TN content, TC: TN and TC: TP ratios were higher than the national average, whereas TN: TP ratio was lower, indicating that carbon, nitrogen, and phosphorus content in the soil was adequate.

### Correlation among components over the life cycle of *Pinus tabuliformis* plantation

According to *Garnier (1998)* theory, there is a positive correlation between the content of an element in the soil and the content of an organ in the plant, and this element is the limiting nutrient. In the present study, the results showed that the carbon content of MF leaves was significantly positively correlated with the soil AP content, whereas the soil TN, TP, and AP content were significantly negatively correlated with the fine roots carbon content. Therefore, the soil AP content in MF was the limiting nutrient for the leaves, and soil TN, TP, and AP were the limiting nutrients for fine roots. Given that soil nutrients have significant effects on fine root biomass and nutrients (*Chen et al., 2016*), it is not surprising that the fine root carbon and nitrogen contents were correlated with the soil carbon and nitrogen contents. There were similar findings, which was consistent with our third hypothesis (*Chen et al., 2018*; *Yuan, Chen & Reich, 2011*). According to the third hypothesis proposed by *Nadelhoffer (2000)* (which is the most well-supported hypothesis), with the improvement of soil nutrient availability, the fine root productivity of trees increases, the life span is shortened, turnover is accelerated, and the fine root biomass decreases. The reduction in the biomass of fine roots of trees can be explained by the optimal allocation and cost-effectiveness theory, that is, the improvement of nutrient availability might reduce the carbon allocation of fine roots. In the *P. tabuliformis* plantation soil in this area, the TN, TP, and AP content in the soil was assumed to be too high, which might reduce the carbon distribution of the fine roots. Among the different organs of *P. tabuliformis* in this study area, only the leaves were significantly correlated with the soil AP content. There was no correlation between branches, trunks, and soil nutrients. These results were consistent with those of *Jiang et al. (2016)*.

## CONCLUSIONS

The influence of *P. tabuliformis* plantation age on the underground element content was greater than that on the carbon content in the aboveground components. The carbon content of the leaves and fine roots, and the TC, TN, and TP content of the soil changed with forest age, whereas the carbon content of the branches and trunks and soil AP did not change significantly with forest age. At the start of the growth, fine roots closer to the trunk had higher carbon content; however, the reverse was true during the later growth stage. Except for soil TP and AP content, the carbon content of fine roots and soil stoichiometry and their ratios decreased with increasing soil depth. The accumulations of soil TC, TN, and AP are long-term process, and TP content has the most evident accumulation trend with an increase in forest age. The analysis showed that the *P. tabuliformis* plantation had a certain effect on increasing the soil nitrogen content, and the carbon, nitrogen, and phosphorus content in the soil was adequate. Soil nitrogen and phosphorus content had a significant effect on the fine roots. In forest management, the soil of *P. tabuliformis* plantation in the eastern mountainous area of Liaoning Province is abundant in nitrogen and phosphorus, so it is unnecessary to apply additional nitrogen and phosphorus fertilizer. These results could provide a useful reference for the management of *P. tabuliformis* plantations and the study of carbon sinks in the mountainous area of eastern Liaoning Province.

## ACKNOWLEDGEMENTS

We are particularly grateful to the following lab members for their help: Z.J., B.Q., and M.L.

### Funding

This research was funded by the National Key R&D Program of China (2017YFD060050102) and the Liaoning Key Research and Development Program (Grant No. 2020JH2/10200033). The funders had no role in study design, data collection and analysis, decision to publish, or preparation of the manuscript.

### Grant Disclosures

The following grant information was disclosed by the authors:
National Key R&D Program of China: 2017YFD060050102.
Liaoning Key Research and Development Program: 2020JH2/10200033.

### Competing Interests

The authors declare there are no competing interests.

### Author Contributions

- Lijiao Wang conceived and designed the experiments, performed the experiments, analyzed the data, prepared figures and/or tables, and approved the final draft.
- Xin Jing conceived and designed the experiments, performed the experiments, prepared figures and/or tables, and approved the final draft.
- Jincheng Han and Lei Yu performed the experiments, prepared figures and/or tables, and approved the final draft.
- Yutao Wang and Ping Liu conceived and designed the experiments, authored or reviewed drafts of the paper, and approved the final draft.

### Field Study Permissions

The following information was supplied relating to field study approvals (i.e., approving body and any reference numbers):

The authorization was granted verbally, onsite clearance was provided by: Yucheng Wang, Dean, College of Forestry, Shenyang Agricultural University; Jianxin Yu, Director, Magu Forest Farm, Forestry Department, Fushun Mining Group Co. Ltd.

### Data Availability

Raw data are available as a Supplementary File.

### Supplemental Information

Supplemental information for this article can be found online at http://dx.doi.org/10.7717/peerj.11873#supplemental-information.

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
