# Peer review of "How C: N: P stoichiometry in soils and carbon distribution in plants respond to forest age in a Pinus tabuliformis plantation in the mountainous area of eastern Liaoning Province, China"

_PeerJ, doi:10.7717/peerj.11873_

## Round 0.1 · original submission · Major Revisions

Dear Dr. Liu and collaborators,

After independent reviews by three reviewers, I believe your manuscript may be accepted for publication in PeerJ as soon as you can modify your text according to the issues raised by the reviewers and prepare a response letter informing them of the changes that were and were not done.

I believe that the necessary changes will be possible to be performed in the period of one and a half months. Therefore, please resubmit up to June 1st, 2021. In case you need additional time, please let me know. Please do not hesitate to resubmit earlier If you can.

Best regards,
Daniel Silva

Reviewer 1 ·

Basic reporting

no comment

Experimental design

correct

Validity of the findings

no comment

Additional comments

How C: N: P Stoichiometry in Soils and Carbon Distribution in Plants Respond to Forest Age in a Pinus tabuliformis Plantation were introduced in this paper. This is a subject of interest to researchers in related fields, but the paper needs a few minor changes before it can be accepted for publication. My detailed comments are as follows:
1. This paper studies the relationship between plant carbon distribution and soil C: N: P stoichiometry in Pinus tabuliformis plantation of five ages from young forest to over-mature forest. This idea is interesting, and the experimental methods adopted are very mature. And in this paper, some important conclusions were drawn, such as “the Pinus tabuliformis plantation had a good nitrogen fixation effect, and the carbon, nitrogen, and phosphorus content in the soil was adequate, and Soil nitrogen and phosphorus content had a significant effect on the fine roots and in forest management, the nitrogen and phosphorus content in soil should be controlled to avoid excessive application of nitrogen and phosphorus fertilizer” and so on.
2. Your result needs more detail. I suggest that you improve the description at lines 170- 172 to provide more justification for your study. And there is an incorrect word on line 212.
3. I suggest simplifying the units of the legend in Figure 2, 3, 4 and 6 to make the figure look cleaner.
4. In the discussion part of this paper, there are three words with dashes and one grammatical error in lines 272 to 277. I suggest you revise it carefully.
I commend the authors for their extensive data set, compiled over many years of detailed fieldwork. In addition, the manuscript is clearly written in professional, unambiguous language. If there is a weakness, it is as I have noted above which should be improved upon before Acceptance.

Reviewer 2 ·

Basic reporting

No comment, see the general and specific comments.

Experimental design

No comment, see the general and specific comments.

Validity of the findings

No comment, see the general and specific comments.

Additional comments

This study deals with C:N:P stoichiometry in soil under five forest ages, those forests are in a Pinus tabuliformis plantation. The sampling site is a pretty good example for the study of ecological stoichiometry in the context of the plant-soil systems along the different forest ages. I have to say I like this study. And I am sure there might be interesting results in the data, howerve, the way the study is described leaves it unclear what the purpose of the data collection in the five forest ages was. In the Discussion section, I can see authors try to discuss and explain why their results are different from other studies, but the discussion needs references, not just mention that the influence is comprehensive or carbon synthesis strategy in leaves differs from those in branches without citation. Also, please do not repeat your results in the Discussion section.

Overall, the subject is relevant. A well-written article, albeit with many grammatical mistakes. I would highly recommend applying a professional language check to correct for many grammatical mistakes and terms used in an incorrect way as well as many articles missing and incorrect singular-plural use.

Annotated reviews are not available for download in order to protect the identity of reviewers who chose to remain anonymous.

Reviewer 3 ·

Basic reporting

Comments to the Authors

The paper by Wang L. et al. addresses the effects of soil C:N:P stoichiometry and plant carbon distribution respond to forest age in a Pinus tabuliformis plantation in the mountainous area. There are many points should be followed.

Abstract:
More numbers should be cited here, such as, you say “Carbon content was highest in the leaves of MAF and NMF and the trunks of YF, MF, and OMF, and was lowest in the branches over the entire life cycle of the aboveground components”. I want to know which is the highest or lowest value. “they had a significant effect on fine roots”…add some numbers will make the Abstract clear for readers if he/she do not see your full paper.

“In forest management, nitrogen and phosphorus content in the soil should be controlled to avoid excessive application of nitrogen and phosphorus fertilizer.” I do not suggest the sentence here, because it is not the major content. You did not summarize your study in the last sentence. I only see many results, but some conclusions should be put here.

Introduction

L41-44, add references.

L47, add point.

L49-52, add references.

L62-63, “and there are few studies on the eastern area of Liaoning Province”. The writing of this sentence is sudden. According to the first half sentence, you can describe the eastern area of Liaoning Province by using some climate attributive, such as temperate zone, etc.

L64, relationships

L64-68, I suggest that the passive sentence can be used here. The references should be put together.

L68 and 70, many studies? Where? List them. Check the whole manuscript.

L109, I suggest that delete the value in each plot, remain the average value in Table 1, while you can list them detailedly in the supplement materials.

Results

L151-180, You say many “there were significant differences between…”, “…showed significant differences”. However, readers need more clear information. If there is no figure, readers can access to information in the writing is important. If you want to express the result is significant, you may use p < 0.05 is OK.

L184, add SE or SD.

L220-221, delete

L227-228, delete

L232, 236, fig. 6

Figs. 5 & 6, I do not suggest the expression. Three point-line is unbelievable (R2= 0.999? 1? Really?). You can add a summary figure by using three colors (a trend line). In the section of materials, you say 360 soil samples? How to calculate? Detailedly. Maybe you can also use all data in the correlations.

Discussion

L287-288, add references.

In the section of 4.3, you should reorganize the data to indicate the conclusion.

Experimental design

no comment

Validity of the findings

no comment

Additional comments

no comment

---

## Round 0.2 · Minor Revisions

Dear Dr. Liu,

I am pleased to inform you that after three reviews, your manuscript is almost accepted for publication. Please check the issues raised by the three reviewers and resubmit your revised manuscript as soon as you can. Do not forget to provide a rebuttal letter informing the reviewer that decided for the minor review of all the changes you performed.

In general, as soon as these issues are corrected, the manuscript is expected to be accepted.

Reviewer 2 ·

Basic reporting

I believe the authors have provided reasonable responses to most of Reviewer 2's questions and concerns to some extent.

Experimental design

I still don’t understand why the authors collected 360 soil samples in 15 permanent sample plots, but mixed the soil sample in the same plot to make only one composite sample.

Validity of the findings

The work is interesting; however, the study scope is also quite narrowly defined.

Additional comments

General comments:
This is a re-review of the manuscript. I believe the authors have provided reasonable responses to most of Reviewer 2's questions and concerns to some extent. The work is interesting; however, the study scope is also quite narrowly defined. The major weakness of the study is the composite sample.

Specific comments:
L278, 281: Please use significantly instead of obviously, if the difference were statistically significant (p < 0.05).
L341, please add the value of “The soil TC: TN ratio in the present study area”.
L354, Add the value of “soil TC: TP ratio” as well as in L356.

---

## Round 0.3 · Minor Revisions

Dear Dr. Liu,

I am pleased to inform you that your manuscript is almost ready to be accepted for publication in PeerJ!

Before I do, the Section Editor noted that "The final bit of the abstract is confusing "and the soil TN, TP and AP 33 were too high, which might reduce the carbon content allocation of fine roots. "

I believe it would be clearer if you edited that to read "...and the soil TN, TP and AP 33 increased, which might reduce the carbon content allocation of fine roots."

Assuming this captures your intended meaning, please edit it and resubmit. Thanks.

---

## Round 0.4 · accepted · Accept

Dear Dr. Liu.

Thank you for your cooperation! I am pleased to inform you that your study has been formally accepted for publication in PeerJ